# A Review of Vision-Laser-Based Civil Infrastructure Inspection and Monitoring

**DOI:** 10.3390/s22155882

**Published:** 2022-08-06

**Authors:** Huixing Zhou, Chongwen Xu, Xiuying Tang, Shun Wang, Zhongyue Zhang

**Affiliations:** 1School of Mechanical-Electronic and Vehicle Engineering, Beijing University of Civil Engineering and Architecture, Beijing 100044, China; 2College of Engineering, China Agricultural University, Beijing 100083, China

**Keywords:** infrastructure inspection and monitoring, non-contact, vision–laser-based, sensor fusion

## Abstract

Structural health and construction security are important problems in civil engineering. Regular infrastructure inspection and monitoring methods are mostly performed manually. Early automatic structural health monitoring techniques were mostly based on contact sensors, which usually are difficult to maintain in complex infrastructure environments. Therefore, non-contact infrastructure inspection and monitoring techniques received increasing interest in recent years, and they are widely used in all aspects of infrastructure life, owing to their convenience and non-destructive properties. This paper provides an overview of vision-based inspection and vision–laser-based monitoring techniques and applications. The inspection part includes image-processing algorithms, object detection, and semantic segmentation. In particular, infrastructure monitoring involves not only visual technologies but also different fusion methods of vision and lasers. Furthermore, the most important challenges for future automatic non-contact inspections and monitoring are discussed and the paper correspondingly concludes with state-of-the-art algorithms and applications to resolve these challenges.

## 1. Introduction

Civil infrastructure, including bridges, roads, dams, tunnels, buildings, etc., greatly improved people’s lives and is also closely related to our safety. Replacing infrastructure or parts of structures would be expensive, labor intensive, and would exceed available financial and human resources; therefore, engineers developed various techniques to ensure the safety and structural integrity of these structures and mitigate financial and life losses from accidents [1]. The traditional method involves visual inspection by trained inspectors, through inspection or monitoring information, combined with relevant decision making criteria to assess the health of civil infrastructure or civil construction [2]. This traditional method has high requirements for inspectors. In addition, owing to the complex environment of infrastructure and facilities, the actions of inspectors are also considerably limited, such as when inspecting tunnels, high infrastructure, and bridges. Therefore, automated intelligent equipment is urgently required to replace inspectors for infrastructure inspection and monitoring. There are some significant differences between inspection and monitoring: inspection focuses on the identification of surface defects in construction materials and structural defects in infrastructure, which is mostly realized by optical cameras and object recognition algorithms; monitoring obtains a quantitative understanding of the current state of the infrastructure by measuring physical quantities, such as defect size, vibration frequency, accelerations, and/or displacement, which is mostly a multi-sensor system.

Owing to the strict requirements of geotechnical structures, defects must be inspected precisely and measured accurately. Over time, intelligent monitoring began using various sensors to replace inspectors. Sensors can be divided into contact and non-contact sensors based on whether they are installed on the infrastructure. Contact sensors, such as force sensors, accelerometers, and temperature sensors, measure physical quantities (such as acceleration, strain, and/or displacement) to quantitatively analyze the current state of a structure. Although such approaches are shown to produce reliable data, they often have limited spatial dimensions and require the installation of dense sensor arrays. Another problem is sensor maintenance once installed. For example, optical fiber sensors, used to measure the strain and temperature information in civil infrastructure, require packaging technique and embedded installation [3,4]. The installation of contact sensors is difficult and time-consuming if only occasional monitoring is required. To address these problems, non-contact inspection and monitoring methods must be developed and tested, reducing human intervention and financial costs, and increasing spatial dimensions. Commonly used non-contact inspection and monitoring sensors include optical sensors, laser sensors, infrared cameras, ultrasonic sensors, Global Navigation Satellite System (GNSS) sensors, and inertial measurement units (IMUs). As common optical sensors, with improvements in image processing technology and the emergence of neural networks, cameras have a vital role in construction, security, and biomedicine. However, cameras also have limitations in that the monocular camera is mostly applied to target detection tasks in two-dimensional (2D) images. Even now, research is conducted on three-dimensional (3D) target detection using monocular cameras; however, these studies rely on a large number of complex features and prior information, and the largest problem is that the 3D detection accuracy of the monocular camera is very low under long distances [5,6,7]. Laser sensors can be divided into single-beam and multi-line laser radars. Single-beam lasers are mostly used to detect translations of a single or a few points in a single direction. Although the measurement accuracy is high, they cannot complete the target detection task alone. For example, a single-beam laser rangefinder can perform single-point ranging to obtain depth information through ToF technology, but due to the lack of 2D plane information and color information, it cannot locate defects and cannot reflect defect properties such as color, area, shape, etc. The largest advantage of multi-line LiDAR is its accuracy. It can identify, classify, and track moving objects using the generated 3D point cloud image, spatial position, and depth information. It has three disadvantages: (1) it is significantly affected by environmental influences, such as rain and fog, (2) it lacks color information, and (3) LiDAR is expensive. Compared with ordinary optical cameras, infrared cameras can provide more reliable information under special conditions, which makes them widely used in the military field. In recent years, with improvements in technology, infrared cameras gradually appeared on the civilian-level consumer market. Ultrasonics is characterized by the strong penetration of liquid and fixed; thus, ultrasonic sensors are commonly used in medicine and underwater monitoring, such as pollution monitoring, earthquake early warning, and auxiliary cruises [8]. Ultrasonic sensors are also used for unmanned ariel vehicle (UAV) landing, obstacle avoidance, ground tracking, etc. The GNSS is primarily responsible for high-precision navigation and positioning functions, and the role of the IMU is to measure the three-axis attitude angle and acceleration of objects.

As mentioned in the previous paragraph, a single sensor can have a certain role but also has limitations: a monocular camera can perform defect inspection in a 2-dimensional plane, but the lack of depth information makes it impossible to quantify defects; a single-beam laser cannot identify the entire object and locate the defects; LiDAR-based monitoring is accurate, but it is expensive, time-consuming, and lacks color information. Therefore, in practical applications, these sensors should be combined to utilize their different functions. This article focuses on vision and the fusion of vision and laser technology. Vision and lasers have various combinations and applications. For example, the integration of cameras and LiDAR is one of the most popular research topics today because it is closely related to autonomous driving. The 2D image information transmitted by the camera is combined with the information provided by the laser. Depth information enables the vehicle to perceive 3D objects, such as people, cars, and the infrastructure around them, as well as transmit the 3D object information to the control unit to enable the vehicle to accurately avoid these obstacles [9,10,11,12]. Currently, many tech companies are investing in autonomous driving, such as Baidu, Huawei, Tesla, and Google. The autonomous driving solutions of various companies differ. In addition to the above-mentioned fusion of cameras and LiDAR, camera–millimeter wave radar combinations are used, such as by Tesla, which uses a multi-camera vision system for 3D object detection. Multi-camera setups are also used to complete some tasks in civil infrastructure, such as the automatic monitoring of defects in 3D space [13,14]. The principle of a multi-camera is an extension of binocular vision, which determines the correspondence between each camera to combine the information transmitted by different cameras to derive depth information in order to form a point cloud. In addition, the multi-sensor combination spurred well-known products, such as total stations and drones. As a measuring instrument, the total station is widely used in many architectural scenes. It is a high-precision measuring instrument integrating light, machine, and electricity, from the earliest theodolite to the present image total station [15,16,17]. For example, Leica’s TS60 realizes the transition from a manual to a fully automatic measurement. The combination of multiple sensors in the total station is a camera–laser rangefinder. In the total image station, two cameras jointly complete the alignment function of the point to be measured, and then the laser rangefinder returns the distance parameters to realize the precise positioning of the point to be measured in the 3D space. UAVs are among the significant inventions of the 21st century. Owing to their power and convenience, drones are widely used in all aspects of life, such as security, construction, and media, and even made significant achievements in the military field [18,19]. A UAV has a high degree of designability. In addition to its aircraft structure, consumers can install various sensors according to their requirements, such as cameras, LiDAR, GNSS, and IMU.

The first paragraph of the introduction described the background of infrastructure inspection and monitoring. The rest of the introduction clarified the motivation for reviewing the non-contact sensors and sensor fusion applications. A thorough review can be found in Section 2 and Section 3, as shown in Figure 1. The former focuses primarily on inspection, whereas the latter focuses on monitoring. The fourth part of the article discusses the challenges of non-contact infrastructure inspection and monitoring. Correspondingly, Section 4 also presents future research and practical engineering applications toward these challenges. The major contents of this paper can be summarized below:Machine vision-based infrastructure inspection, especially semantic segmentation.Infrastructure monitoring and a quantitative understanding of the current state of the infrastructure.Vision–laser fusion technologies and their applications.The challenges and ongoing works toward automated non-contact infrastructure inspection and monitoring.

## 2. Vision-Based Infrastructure Inspection

Infrastructure inspection is an application of object detection technology in the field of construction. Relevant data are collected through various types of sensors, such as images, laser signals, and ultrasonic signals, and these signals are further processed to separate different characteristics to determine the state of the infrastructure. In recent years, the continuous development of computer vision technology enabled the application of visual inspection in various engineering fields. Recent research on automatic vision-based inspection is generally divided into two steps: (1) acquiring image data by setting up static cameras or flying drones [20,21]; (2) processing image data by computer vision technology to complete defect and structure inspection. There are some challenges in data acquisition: (1) a single static camera is limited by its field of view; (2) the flight path of camera-equipped drones heavily affects inspection results. These challenges are discussed in Section 3 and Section 4. Section 2 focuses on data processing techniques and introduces vision-based infrastructure inspection from three perspectives. The first part presents the types of most defects and different heuristic methods for damage detection using image data, and the other sections review the application of artificial intelligence technology in infrastructure inspection, which is divided into two parts: object detection and semantic segmentation.

### 2.1. Image Processing Algorithms

Image processing algorithms primarily include grayscale transformation, filtering, morphology, feature detection, and region segmentation. Typical crack detection is accomplished through a combination of these techniques. Yiyang et al. [22] proposed a traditional image processing method to identify glass cracks. First, the collected images are pre-processed and corresponding filters are selected for different noises to reduce the interference. The second step is to sharpen the image. In the process of denoising, sharpening, and edge detection, the output results are generated by convolving the input image with the filter. Therefore, the choice of filter is particularly important. Salman et al. [23] used the Gabor filter for road crack detection. They indicated that the Gabor filter can extract crack features at different scales and different directions, similar to human visual perception. Experiments showed that the proposed method has 95% accuracy. After filtering, the image is frequently binarized to better segment the crack area. Li et al. [24] used the Ostu method to perform threshold segmentation on an image. Based on the normalized histogram, the method counted each gray value class and calculated the inter-class variance. The largest inter-class variance was used as the global threshold for image segmentation. Crack detection using image binarization is challenging because of different parameter choices and methods. Kim et al. [25] compared the performance of five common binarization methods in concrete crack detection in terms of crack length, width, and calculation time. The final experimental results indicate that most of the methods can effectively identify cracks, and combining multiple methods for identification has significant potential. The effectiveness of a crack detection method is measured by both the accuracy rate and computation time, particularly when a large number of images must be inspected. Yamaguchi et al. [26] proposed a rapid crack identification method. This method sets the initial gray value threshold, continuously compares the pixel value in the window and the threshold to determine the segmentation area, and updates the threshold through the acceleration parameter; it terminates when the window size is equal to the maximum value. The author proved the effectiveness of the algorithm in identifying cracks through experiments, and the calculation time is highly correlated with the initial threshold setting. In an actual engineering environment, the identification of cracks must also eliminate disturbances, such as shadows, rain, and fog. Zou et al. [27] proposed CrackTree, a fully automated method for identifying road cracks. This method can solve the problems of low contrast and shadows of cracked images. In addition to shadow occlusion, some infrastructure occlusion is used. Yeum et al. [28] conducted multi-angle shooting to solve such problems. Because two objects are present in the captured images (rivets and cracks), the two objects must be classified first. After classification, denoising, edge detection, dilation, etc., are used to segment the crack area. After crack identification, quantitative data are often required for further analysis of the entire road or infrastructure. While quantitative crack identification requires pixel-wise features, Zhu et al. [29] proposed a new crack attribute retrieval method. After identifying the crack, the crack feature is skeletonized, and then distance transformation is used to calculate the crack width, length, direction, and other attributes. Nishikawaet et al. [30] designed a filter based on genetic programming to identify cracks and reused the filter to remove noise around cracks after region segmentation. Finally, the crack width can be quantified from the spatial derivative of the luminance pattern.

The corrosion of infrastructure surfaces is also frequently discussed under infrastructure health. Medeiros et al. [31] proposed a corrosion feature descriptor to separate corroded and uncorroded regions. Under the separated region of interest (ROI) area, the surface features are separated using a gray-level co-occurrence matrix (GLCM), and the color features are obtained using statistics from HSI data. These two features form the corrosion descriptor. Finally, Fisher’s linear discriminant analysis is used to distinguish the descriptors by dividing the corrosion area. Jahanshahi et al. [32] used wavelet transform to identify corrosion features. They first evaluated several parameters that can affect the performance of the wavelet transform and then proposed the use of depth perception to complete corrosion identification. The limitation of this method is that the object distance must be fixed and the intrinsic parameters of the camera must be calibrated. Shen et al. [33] developed a new rust defect identification method for noise and non-uniform illumination. This method uses Fourier transform to determine surface defect pictures and then applies color image processing to identify rusted color pictures. Combining the results of the two enables rust defect identification in a specific environment, and the calculation time is short, which can be used as a real-time steel inspection method.

In addition to cracks and surface corrosion, other defects can be detected in infrastructure health. For example, Li et al. [34] designed a real-time rail inspection system for rail defects. Because the image acquisition device is fixed at the bottom of the train, the ROI can be separated by the actual rail size, shooting height, and high gray value characteristics of the rail; subsequently, local normalization can be used to improve contrast, and finally, defects can be identified using the defect localization based on a projection profile. For road inspection, Koch et al. [35] proposed the use of histogram shape-based thresholding, morphological refinement, and elliptic regression to separate defect areas and compare them with non-defect areas to automatically identify large pits on asphalt roads. This section introduced the application of various image processing algorithms in infrastructure inspection, and a summary is shown in Table 1, but we can observe that these involve handcrafted problems, such as filter parameter selection and binarization threshold setting. Although the improved algorithm achieves fully automated defect inspection, robustness cannot be guaranteed, owing to the problem of adaptive parameters. With the emergence of machine learning algorithms, this problem is gradually solved, and the related algorithm principles and applications are introduced in the remaining sections.

### 2.2. Object Detection

Object detection aims to distinguish different objects and accurately estimate the position and concept of the object in the image [36]. As mentioned in Section 2.1, for improved robustness of algorithms in identifying infrastructure inspections, machine learning algorithms are gradually replacing image processing algorithms. Commonly used machine learning algorithms include the support vector machine (SVM), K-nearest neighbors (K-NNs), naive Bayes, decision tree, random forest, and neural networks. Gibert et al. [37] utilized histogram of gradient orientation (HOG) features and a linear SVM classifier to inspect various types of defects in railway tracks. In general, machine learning algorithms must provide training sets to train models, which can be divided into supervised and unsupervised learning according to whether the training sets are labeled. For example, the SVM is a supervised classification algorithm. The advantage of unsupervised learning, such as clustering algorithms, is that it saves time in labeling data. Feng et al. [38] proposed a fully automated rail inspection system based on latent Dirichlet allocation (a probabilistic clustering algorithm). The classification model structural topic model (STM) in the article is an extension of latent Dirichlet allocation (LDA), and it solves the limitations of LDA by ignoring the spatial relationship between visual words. At the end of the article, the classification results of STM, SVM, boosted tree, and neural networks were compared. Inkoom et al. [39] evaluated the performance of partitioning, bootstrap forest, boosted trees, naïve Bayes, and K-NNs in road crack detection. The multi-classifier neural network algorithm, as a popular research topic today, has advantages, such as robustness, strong adaptability, and complex nonlinearity. The neural network layers can be divided into shallow and deep neural networks. Boltzman machines, proposed by Hinton et al. [40] in 1985, are shallow neural networks with one visible layer and one hidden layer. Xu et al. [41] developed a method for detecting infrastructure surface cracks based on restricted Boltzmann machines.

When the number of hidden layers increases, the shallow neural network becomes a deep neural network. Compared with shallow neural networks, deep neural networks can detect smaller details, whether they are image or sound data. Deep neural networks also have a significant role in infrastructure health, and Bao et al. [42] attempted to visualize time-domain signal data and then used the visualized data to train deep neural networks to detect infrastructure anomalies. In visual-based infrastructure inspection, the original data are generally image data, which means the original neural network has too many features. A large number of features can easily cause overfitting, particularly because the image resolution is improved today. The emergence of convolutional neural networks (CNNs) solved this problem ([43,44,45,46] describe the development history of CNNs). Owing to the improvement of hardware computing power, CNNs are widely used in various engineering fields. Cha et al. [47] used a CNN to identify surface cracks with 98% accuracy. Yeum et al. [48] used a CNN for post-event infrastructure reconnaissance to detect collapse problems. Refs. [49,50,51] compared the performance of CNNs with other algorithms in crack recognition. Kim et al. [49] evaluated the performance of CNN-based and SURF-based methods in identifying cracks and non-cracks. Chen et al. [50] proposed an NB-CNN combining naive Bayes and a CNN, and compared it with LBP-SVM. Zhang et al. [51] compared the performance of the CNN, SVM, and boosting in detecting cracks, and confirmed that CNN can better separate cracks from the road background. Ali et al. [52] investigated the performance of popular machine learning algorithms, such as KNN, logistic regression, SVM, random forest, naive Bayes, and CNN, in crack detection. The final results show that the KNN, random forest, and neural network performed better, but the neural network consumed the most time. This result shows that for a specific detection task, engineers should comprehensively consider factors such as accuracy and time cost to select an appropriate machine learning algorithm. For defects, such as cracks and corrosion, a bounding box cannot accurately express the defect location. Kim et al. [53] proposed an automatic crack detection method based on AlexNet, where the final crack area was displayed in an irregular shape. Figure 2 shows the architecture of AlexNet. Atha et al. [54] used different CNN architectures to complete corrosion detection and compared their performance. The input image is captured by sliding windows of different sizes to make the detection results more accurate. However, Atha’s method has limitations: (1) the sliding window must scan the entire image, which is inefficient, and (2) only a single defect can be identified. To solve the above problems, Cha et al. [55] provided an inspection method based on the faster RCNN to identify various types of defects. While identifying five types of defects with high accuracy, it also achieved the detection speed of 0.03 s for a single image. A summary of object detection is presented in Table 2.

### 2.3. Semantic Segmentation

The end of Section 2.2 mentions that, to achieve the accuracy of corrosion detection, a smaller sliding window can be selected to cover the entire corrosion area. However, the smallest window size in Ref. [54] was 32×32. To achieve pixel-wise object detection, scholars proposed the concept of semantic segmentation. Semantic segmentation is a high-level task that facilitates complete scene understanding. It can classify every pixel in an image and is widely used in medical imaging and autonomous driving. Similarly, infrastructure inspections have related applications. Hoskere et al. [56] developed a defect detection architecture based on a multi-scale pixel-level CNN. In the architecture, the Gaussian pyramid is first used to generate multi-scale pictures; then, the segmenter used is ResNet23, and VGG19_reduced is used as the classifier to complete the identification of six types of defects. However, the results indicate that the target segmentation effect of this framework is poor, which was also a common problem encountered in early semantic segmentation, until the fully convolutional network (FCN) was founded by Long et al. [57]. For the same target detection task, Hoskere et al. [58] proposed the condition-aware model based on the FCN, and the detection results of multi-scale pixel-level CNN and FCN were compared to demonstrate the effectiveness of the FCN. In terms of rail detection, the FCN also exhibited good performance. Giben et al. [59] designed a rail material classification model based on the FCN, which distinguished ballast, wood, lubricator, rail, fastener, and various forms of concrete. However, the FCN also has limitations. Its classification results are not sufficiently accurate, and it is not sensitive to details. It does not consider the relationship between pixels and loses some spatial information. This is because the decoder part of the FCN is very simple and only one deconvolution operation is performed. Therefore, many researchers proposed more complex decoder structures to achieve more accurate semantic segmentation. Islam et al. [60] designed a decoder structure composed of multiple upsampling layers, deconvolution layers, and full convolution layers based on VGGNet, and achieved a 91.3% accuracy in concrete crack detection (Figure 3). To achieve accurate localization of all defects, Liang et al. [61] applied the encoder–decoder structure of the FCN, and Bayesian optimization was used to determine the value of the hyperparameters to improve the robustness of the model.

In summary, an excellent decoder structure can achieve better performance in semantic segmentation tasks. However, there is insufficient information regarding this application. The direct deconvolution of the deep layer cannot provide detailed information. The U-Net proposed by Ronneberger et al. [62] provides local details by stacking more skip connections. The U-Net architecture is illustrated in Figure 4. Similarly, the U-Net is widely used in infrastructure inspections. Enshaei et al. [63] efficiently inspected textured surface defects using the U-Net. Pan et al. [64] implemented an improved U-Net model to inspect sewer pipes. Feature reuse and an attention mechanism were added to the original skip connection to enhance feature extraction, and multiple defect detection in the pipeline was realized with 32 pictures per second. As a semantic segmentation model, U-Net can cooperate with other object detection networks to achieve a more accurate detection. Wei et al. [65] proposed a hierarchical semantic segmentation strategy to identify highway marking defects. A faster RCNN was used as the target detection network, and the detected road signs were sent to U-Net for ratio defect detection. The semantic segmentation of 3D infrastructures can also be achieved. A 3D metric concrete inspection system designed by Yang consisted of three parts: simultaneous localization and mapping (SLAM) for positioning association, a U-Net-based deep neural network for defect segmentation, and a Bayesian filter for 3D semantic fusion [66]. Both FCN and U-Net have a problem in that max pooling will cause the loss of position information. To solve this problem, Badrinarayanan et al. [67] proposed SegNet with coordinate pooling. Zhang et al. [68] adopted an adaptive sliding window to obtain a dataset based on SegNet, then used NMIPS to retain the image patches with significant local edge textures to save overall time, and finally synthesized the semantic segmentation results through CAOPF to project onto the original image.

As mentioned in Section 2.2, machine learning algorithms are divided into supervised and unsupervised learning. When preparing for model training, supervised learning requires a large amount of time to label the original data. This disadvantage is particularly apparent in semantic segmentation. Therefore, many scholars proposed semi-supervised learning models that reduce time consumption while ensuring accuracy. Wang et al. [69] proposed a U-Net-based EfficientU-Net model to achieve an accuracy of 83.21% with 60% annotation in a surface crack detection task. A summary of semantic segmentation is presented in Table 3.

### 2.4. Summary

Image processing algorithms, object detection, and semantic segmentation are vision-based infrastructure algorithms. The image processing algorithm attempts to use a few adaptive parameters to automatically detect infrastructure defects. To realize multi-parameter and multiclass defect classification, the object detection algorithms, such as CNN, SVM, and K-NNs, are used in infrastructure inspection. Semantic segmentation instead achieves pixel-wise defects detection. However, these three vision-based technologies lack the quantitative results that researchers require to analyze a structure and perform a condition assessment. The next section investigates the different infrastructure defect measurement technologies.

## 3. Vision–Laser-Based Infrastructure Monitoring

For non-contact infrastructure inspections, the previous section highlights the application of vision technology. Owing to the rapid development of neural networks and deep learning, vision technology has an important role in infrastructure inspection. From the identification of various defects using image processing algorithms to object detection and semantic segmentation, various infrastructure defects are accurately inspected at the pixel level. However, defect detection and classification results cannot be quantified, resulting in a lack of data support for further analysis of the infrastructure. For example, for road cracks, the actual sizes are not only relevant to road safety, but they determine the maintenance time. In addition to infrastructure health, monitoring technology is widely used in the construction process. This includes material quality monitoring, foundation pit deformation monitoring, and construction quality assessment. However, infrastructure monitoring was originally completed manually or with contact sensors. As mentioned in the first section, contact sensors have problems, such as complicated deployment and maintenance. Therefore, non-contact monitoring is a research problem and trend in civil infrastructure. In this section, the application of visual measurement technology in infrastructure monitoring is introduced, and then the applications and advantages of the fusion technology of vision and laser in infrastructure monitoring are discussed.

### 3.1. Vision-Based Monitoring

#### 3.1.1. DIC

As the title of Ref. [70]: image correlation for shape, motion, and deformation measurements. Digital image correlation (DIC) technology is a visual measurement technology. Its principle is to divide the ROI of two digital images before and after deformation into several sub-regions and obtain the displacement of the corresponding sub-regions through correlation calculations. The deformation information of an entire field can be obtained. DIC technology can measure the target deformation and strain, and has the advantages of full-field measurement, strong interference ability, and high measurement accuracy. DIC algorithms can also apply different post-processing steps to compute 2D in-plane strain fields (2D-DIC), out-of-plane displacement and strain fields (3D-DIC), and volumetric measurements (VDIC).

DIC can be used to measure the deformation and strain of various infrastructures such as bridges and beams. To solve the measurement error caused by the out-of-plane motion of DIC when measuring 2D strain, Hoult et al. [71] used five methods to reduce the error and confirmed the effectiveness of three of them, indicating that DIC can replace the conventional strain possibility of gages. In terms of beam condition monitoring, Dutton et al. [72] used DIC to monitor the curvature of beams. In addition to the bending rate, the fatigue behavior of reinforced concrete beams must also be considered. Mahal et al. [73] collected various types of information, such as beam deflection and curvature, as well as crack width and height through DIC. Thus, DIC is highly effective for bridge monitoring. Yoneyama et al. [74] used DIC to measure the deflection of a bridge in a load-bearing state. To measure the complete bridge body in the field experiment, they set up a camera at each end of the bridge to generate a contour map to analyze the reflection distribution of the bridge girder. DIC not only has 2D-DIC but also 3D-DIC. Chen et al. [75] achieved full-field 3D measurements using multiple cameras. In the 3D-DIC measurement system, multiple cameras were mapped to a unified coordinate system through calibration, and the contour of the object to be measured was generated to obtain deformation information. Similarly, Helfrick et al. [76] proposed 3D-DIC-based shape and displacement monitoring of vibrating structures. Ghorbani et al. [77] used 3D-DIC to complete full-field deformation measurement and crack mapping of confined masonry walls. Compared with manually drawn maps, crack mapping based on 3D-DIC maximum principal strain maps can express the wall state in more detail and reduce human error. An important point of the DIC method is that it does not require expensive sensors, and even a mobile phone can use DIC for complete displacement monitoring. Wang et al. [78] proposed a smartphone-based 3D structural displacement monitoring system. The effectiveness of the system was verified through dynamic and static experiments.

#### 3.1.2. MVS and SFM

Multiview stereo photogrammetry (MVS) is a method of infrastructure monitoring that utilizes multiple cameras. Compared with DIC, MVS must know the camera pose as well as the intrinsic and extrinsic parameters of the camera, and the mapping matrix between multiple cameras must be obtained through camera calibration. A commonly used calibration method is the checkerboard calibration method proposed by Zhang et al. [79]. The checkerboard was used as a target to measure the displacement of bridge piers in Ref. [80]. Del Sal et al. [81] proposed a multi-camera-based structural vibration measurement system. After the fixed multi-camera completed the camera calibration, the landmark of the area to be measured was identified to achieve an accurate measurement. The method described in Ref. [81] has certain flaws, and the monitoring systems require marker points. However, in actual infrastructure conditions, arranging marker points is equivalent to arranging contact sensors for measurement; thus, a feature point is required to replace artificial markers. Lowe et al. [82] proposed that the SIFT feature can effectively achieve this objective. Shan et al. [83] measured the cracks on concrete surfaces using the SITF feature and a calibrated stereo vision system.

As mentioned for MVS, a checkerboard calibration method is required to determine the camera pose. The checkerboard calibration method is also a method that relies on feature matching. Similarly, as a scale-invariant feature, the SIFT feature can replace the calibration board to estimate the pose of the camera and generate a 3D point cloud. This is the principle of structure from motion (SFM). Liu et al. [84] evaluated cracks using a 3D point cloud formed using SFM. To measure the thickness of surface cracks, Jahanshahi et al. [85] proposed a new crack monitoring method that realizes crack identification and quantification at any distance, focal length, and resolution through the incorporation of depth perception. To establish depth perception, they used the SFM 3D structure reconstruction method based on the SITF feature points. Torok et al. [86] designed a ground robot for post-earthquake infrastructure evaluation of large-scale cracks. The camera was installed on the robot platform to monitor post-earthquake infrastructure structural changes using SIFT feature points and SFM 3D reconstruction methods. Generally, when the feature points are insufficient or the environment is complex, the point cloud generated by SFM is sparse. To establish a dense point cloud for accurate 3D reconstruction, Parente et al. [87] proposed multi-camera monitoring based on SFM-MVS. The system also uses the SIFT feature to achieve image matching to create a dense 3D point cloud to monitor infrastructure surface changes. As the most effective point feature, SIFT has the disadvantage of being time-consuming, and SIFT-based 3D reconstruction overcomes this limitation. Therefore, to improve efficiency, Bay et al. [88] proposed SURF feature points, which are faster scale-invariant features. Özcan et al. [89] used dense image matching (DIM) based on the SURF feature to monitor the roughness of concrete structures. Experiments showed that, compared with SIFT, SURF guarantees shorter time consumption with the same accuracy rate. Compared with SFM, DIM realizes the matching of each pixel point to ensure monitoring accuracy. Figure 5 shows the results of the sparse and dense 3D reconstruction. As a commonly used vision-based 3D reconstruction method, SFM is often compared with other 3D reconstruction techniques to verify its accuracy. Refs. [90,91,92] compared SFM and laser 3D reconstruction and monitoring accuracy. The results of manual laser scanning, ground-penetrating radar scanning, LiDAR 3D reconstruction, and SFM 3D reconstruction were compared. The results show that although SFM optimized the matching error in different ways to form dense point clouds, the accuracy and density of the point clouds were still lower than those of laser 3D reconstruction. A summary of the vision-based infrastructure monitoring is presented in Table 4.

### 3.2. Laser–Vision Fusion

The vision-based infrastructure monitoring technologies mentioned in Section 3.1 (DIC, MVS, and SFM) have their limitations. DIC is a static measurement technology that requires a strict experimental layout and a measurement environment. MVS must calibrate the camera pose and arrange the control points in advance. SFM mostly completes monitoring and 3D reconstruction through SIFT feature point matching. Although SIFT feature points are effective, they still cause matching errors in complex architectural environments, forming sparse 3D point clouds. Laser 3D reconstruction technology is discussed at the end of Section 3.1.2. Even if the accuracy rate is high, defects remain. Refs. [9,10,11,12] described vision and laser technologies used in autonomous driving, and the corresponding fusion technology can also be used in infrastructure monitoring to achieve high-precision displacement, vibration, and defect monitoring.

The most necessary information when quantifying defects is depth information. When the depth of any object or even each pixel in a picture is known, the 3D information of the object can be recovered to measure the defects and realize infrastructure monitoring. Lasers and vision can restore the depth of information through different fusion methods. In the three subsections of this section, three fusion methods of laser and vision fusion and their applications in infrastructure monitoring are described.

#### 3.2.1. Laser Range Vision

Single-point displacement monitoring tasks are often involved in infrastructure monitoring, such as bridge health and slight deformation of foundation pits. This type of problem can be transformed into transformation monitoring of the point to be measured in a 3D space in large-scale space. Because the target to be measured is small, and the monitoring distance is long, a combination of long-distance laser ranging and visual detection technology is required. The image total station is a typical combination of laser ranging and vision, and it has a significant role in the field of infrastructure monitoring.

The traditional total station consists of telescopes, eyepieces, and laser transmitters and requires manual alignment during measurement. Therefore, the traditional total station can only measure one point at a time, and the measurement accuracy and efficiency are affected by the experience and technical level of the surveyor [93]. To solve this problem, Walser et al. [15] developed an image total station. The image-assisted total station primarily uses two types of sensors, a camera, and laser, as automatic measurement equipment [94] (Figure 6). The laser sensor is primarily used for ranging, and the cameras are divided into wide-angle and telescope cameras. The fixed focal length of the wide-angle camera primarily searches for the target to be measured through object detection. A telescopic camera is installed coaxial to the collimation axis and subjected to optical magnification and variable focus. Benefiting from the magnification of the telescope, the resolution of the image is extremely high, and high measurement accuracy of the point to be measured is achieved.

Wagner et al. designed three monitoring systems based on image total stations in Refs. [95,96,97]. (1) In Ref. [95], they proposed using an image total station as a video rangefinder for bridge monitoring. The system monitored bridge motions and oscillation frequencies through actual measurements. (2) In [96], they designed a new long-distance monitoring system. The system consisted of two image stations to establish a 3D total station monitoring network, which could be primarily used for long-distance 3D deformation monitoring of foundation pits. (3) In [97], they developed a new approach for geo-monitoring, using modern total stations and RGB+D images. The RGB+D camera provided color and depth images for the entire system. Multiple images should be stitched to solve the problem of a single image having a small field of view and covering fewer areas. The traditional splicing method is primarily based on feature point matching, and the matching results are poor. Therefore, this system uses the total station to unify the coordinates of multiple images to achieve perfect image matching results. Finally, the stitched images are analyzed to determine whether the area to be measured is deformed.

In addition to the image total station, some related infrastructure-monitoring applications combine laser ranging and vision. Vasileva et al. [98] proposed a monitoring system similar to the image total station, but different from the total station. The laser and camera of this system were not coaxial, and a heating device was added to monitor the infrastructure displacement at a low temperature. Zhang et al. [99] employed an automatic crack detection system to monitor the safety of subway tracks. This system was also equipped with a laser ranging sensor to measure the distance from the camera to the surface.

#### 3.2.2. Laser Structured Light

Laser structured light is an active optical measurement technique that projects a laser point or line through an emitter onto the surface of the object to be measured, and the image is acquired by an image sensor. The 3D coordinates of the object are calculated through systematic geometric relationships. Myung et al. [100] developed a paired structured light module that consists of cheap cameras and point lasers to measure the accurate displacement between any two locations on the structure. Compared with point lasers that provide distance information, a line laser is more of a “tool.” The camera captures the distribution of the line laser on the plane to be measured, which is further analyzed using visual methods to obtain quantitative features. This is similar to the depth information restoration of structured light. Another difference is that laser ranging is often used for infrastructure monitoring at medium and long distances. Line lasers are more of a close-up application and are mostly used to monitor infrastructure surface defects.

In the initial design of the expressway, to know the relationship between the friction force provided by the asphalt and the depth of different types of asphalt, Ding et al. [101] used a line laser to scan an entire asphalt sample in order to generate a 3D laser image and segmented the test area to obtain the depth of the surface of different asphalt samples. To monitor highway defects, Arezoumand et al. [102] combined a line laser and high-speed cameras to complete the rutting measurement of the highway. When the line laser swept over the road surface, the camera recorded the corresponding pictures. In the data processing stage, a filtering operation was used to segment the laser ROI area, and then K-means clustering was used to measure the rut depth. Changes in the structure of the highway, owing to the stress due to carrying vehicles over a long period, affect the resistance of the highway and the comfort and safety of consumers. For continuous observation and accurate analysis of the road surface, Vilaça et al. [103] proposed a 3D surface scanning device that consists of a line laser and two cameras, and it was used to calculate the texture depth and texture profile level of the area to be measured in order to analyze the state of highways. In terms of accurate monitoring of underground rails, Zhan et al. [104] proposed a vision system combining multiple cameras and structured light, and they built an overall measurement model by calibrating multiple cameras and structured light cameras to achieve high-precision 3D in field experiments. Point cloud restoration provides data support for the real-time monitoring and maintenance of tunnels.

#### 3.2.3. LiDAR Vision

LiDAR is a measurement technology that emits a pulsed laser beam to a target and then measures the arrival time, strength, and other parameters of the reflected signal to determine the distance, orientation, motion state, and surface optical properties of the target. Because of its accuracy, LiDAR has a significant role in surveying and mapping, agriculture, construction, and environmental monitoring [105,106,107,108,109]. However, LiDAR has its flaws, as mentioned in the first section: (1) the more accurate multi-line LiDAR is expensive; (2) it is affected by the environment, such as rainy days, foggy days, and objects to be measured with different reflectivities; and (3) the generated point cloud data cannot reflect color information. Therefore, the fusion of LiDAR and vision became a research topic in recent years. It can also be used for infrastructure monitoring. Ref. [110] mentioned that, although LiDAR and vision fusion can have their respective advantages, they require a difficult calibration step (Figure 7). Refs. [111,112,113,114] described manual, automatic, and real-time calibration methods based on CNN. After the calibration is completed, the LiDAR vision can be transformed to unify the coordinates of the LiDAR and the camera. For actual scenarios, Omidalizarandi et al. [115] proposed a robust extrinsic parameter calibration method, which integrates cameras and laser scanning equipment to achieve high-precision infrastructure deformation monitoring.

The LiDAR vision system has the advantages of high accuracy and good mobility for infrastructure monitoring. The visual measurement techniques mentioned in Section 3.1, such as MVS and SFM, can be used to monitor large infrastructure; however, they can only form sparse point clouds in most scenarios and require high image quality and resolution. LiDAR can compensate for the limitations of image processing in infrastructural monitoring. For example, in terms of crack monitoring, Valença et al. [116] evaluated cracks in concrete bridges using a LiDAR vision system, and the geometric information provided by TLS complemented the limitations of image processing. Similarly, Rabah et al. [117] mapped the detected crack pixel coordinates to a global coordinate system after identifying cracks, and the measurement errors in the three directions were 30, 16, and 14 mm. In terms of road pit monitoring, Kang et al. [118] used two 2D LiDAR sensors and one camera to complete pit monitoring under laboratory conditions and identified rectangular pits through image processing algorithms, such as denoising, edge detection, and target segmentation. LiDAR was used to obtain the camera’s shooting distance and angle information for accurate quantification. In an actual engineering environment, Wu et al. [119] used mobile mapping sensors to monitor the 26.4 km Shanghai highway. LiDAR–vision fusion systems were also used in other monitoring tasks. Kashani et al. [120] proposed a monitoring method based on ground-based LiDAR, which used the grayscale and color information provided by vision and LiDAR point cloud information as the input data of the clustering algorithm to detect roof-covering defects. There are various types of surface defects, such as corrosion, deformation, cracks, and pits. To identify and quantify diverse surface defects, Erkal et al. [121] developed a surface property monitoring method using LiDAR vision, which is primarily used to provide color data as fourth dimension information. In terms of subway safety monitoring, Zhangyu et al. [122] fused camera and LiDAR data to detect small obstacles and vehicles. Vision technology achieved pixel-level ROI area separation, and LiDAR data were used to estimate the distance of vehicles ahead and detect small obstacles. As mentioned earlier, most of the sparse point clouds formed by the 3D reconstruction of large infrastructures using only visual technology have significant errors. Zhen et al. [123] proposed a LiDAR–camera fusion method for accurate dense 3D reconstruction methods (Figure 8). This method uses the iterative closest point (ICP) algorithm to solve point cloud matching and bundle adjustment problems, and realizes mm-level 3D reconstruction for large structures, such as bridges, columns, and squares.

With the development of UAV technology, UAV monitoring systems equipped with multiple sensors are gradually becoming a mainstream monitoring method. Refs. [124] and [125] introduced a bridge crack monitoring system based on UAV radar vision. Jung et al. [124] used LiDAR, cameras, and other sensors, such as GPS and IMU, to plan a UAV route better. The position and direction information of the drone obtained simultaneously made the processing of point cloud and image data smoother and more accurate. A monitoring error of at least 9 mm was achieved. Yan et al. [125] mounted a high-resolution camera and Velodyne VLP-16 LiDAR scanner on a drone. First, a camera was used to lock the ROI that required laser scanning, and then LiDAR was used to obtain depth information to complete the quantification of bridge cracks. Özaslan et al. [126] used a microdrone to monitor dark featureless dam penstocks. The drone was equipped with four cameras, which enabled the system to complete 360-degree splicing of the pipeline and locate pipeline cracks and rusty spots from the spliced images. A multi-LiDAR system was used to reduce errors to obtain accurate distance information. UAVs equipped with LiDAR vision systems also have a significant role in the 3D reconstruction of infrastructure in large areas. Li et al. [127] used pre-earthquake LiDAR data and high-resolution images to reconstruct seismic areas. Subsequently, each rooftop patch of the 3D model was locked, and the LiDAR data transformation in this area was compared before and after the earthquake to quantitatively analyze the degree of damage. Hirose et al. [128] proposed a mobile post-earthquake monitoring system without preinformation. The camera was primarily used to identify landmarks in the environment to provide the position and direction information of the UAV. Kalman and particle filters were used to optimize the flight trajectory in order to realize accurate monitoring in an unknown environment. High-precision monitoring can also be used for the 3D recording of cultural relics. Murtiyoso et al. [129] combined terrestrial laser scanners, DSLR cameras, and drones to achieve a multi-scale recording of cultural relics in detail. After completing the data collection, the final step of the system would georeference the close-range image and laser-scanned point cloud to the same system using area markers. Figure 9 shows the results of the 3D documentation of cultural relics. A summary of Section 3.2 is given in Table 5.

## 4. Challenges of Non-Contact Monitoring

According to the sensor-related technologies and applications described above, optical and laser sensors are widely used in the field of infrastructure inspection and monitoring. However, some problems that impede the practical application of non-contact sensors remain.

### 4.1. Model Training Requires Large Amounts of Data

In the application of infrastructure inspection based on computer vision, commonly used models such as CNNs, RCNNs, and FCNs require a large amount of image data for training; thus, the labeling of data before training requires a significant amount of time. Additionally, the larger the image dataset, the less conducive it is to data transmission and storage. Therefore, it is very important to use a small number of training sets to build effective machine learning models. Based on this, Bertinetto et al. [130] proposed one-shot learning. Finn et al. [131] developed a metalearning approach. Fink et al. [132] described a framework for learning an object classifier using a single example. Kim et al. [133] proposed a few-shot learning approach that can successfully learn and detect new construction objects when only a small amount of training data is provided.

### 4.2. Model Transferability

The use of deep learning to complete inspection and measurement tasks frequently encounters two problems: (1) the large amount of data mentioned in the previous subsection and (2) the heavy reliance on the hardware configuration. When encountering a new recognition or prediction task in a project, researchers hope that the new dataset can be applied to the already trained model to obtain the correct recognition result. Therefore, Yosinski et al. [134] first proposed the concept of transfer learning. For defect detection, Yang et al. [135] developed an end-to-end transfer learning crack detection method based on VGG16, which combined sample, model, and parameter knowledge. Aliyari et al. [136] used various CNN networks to inspect bridges. The authors discussed the importance of selecting an appropriate amount of transfer learning and a suitable training set and evaluated the performance of CNN models through transfer learning in the presence of natural noise.

### 4.3. Noise Influence

Environmental impact problems often occur when using sensors for inspection and monitoring in the real world. Optical sensors, such as cameras, can be easily affected by light. Light that is too strong or insufficient affects the image quality. Shadows occlude key areas, and feature points are common problems in image processing. The infrastructure environment in which the sensor is placed is complex. In addition, a few denoising algorithms are available to resolve this influence. He et al. [137] proposed a dehazing method based on a dark channel prior. Yan et al. [138] proposed a self-alignment network to solve the problems of rain streaks and rain accumulation. Compared with rain removal and fog removal, more studies were conducted on solving shadow occlusion. Jianyong et al. [139] developed a vehicle shadow removal method based on texture autocorrelation and integer wavelet transform. Finlayson et al. [140] presented a shadow removal method based on entropy minimization to determine the invariant direction.

### 4.4. Expensive Sensors

As mentioned in Section 1, LiDAR is an expensive sensor. Designing a low-cost system to complete high-precision monitoring without LiDAR is a significant challenge. Lydon et al. [141] designed a contactless, low-cost vision-based displacement measurement system that used a lower-priced action camera GoPro with a 25–135 mm F1.8 C-mount lens. Kuhn et al. [142] designed an underwater infrastructure inspection device. This device used low-cost sensors to control the direction and position of the entire device while checking underwater structures, such as cables and water pipes. Lei et al. [143] proposed a new low-cost UAV for crack inspection. Zhang et al. [144] presented a novel TLS to realize the low-cost method for indoor quality inspection, which consists of four 1-D laser sensors and one camera (Figure 10). Apparently, the replacement of expensive sensors, such as LiDAR, with cheap sensors, such as cameras and 1-D laser sensors, is the key to designing low-cost systems.

### 4.5. Decision-Making Problem

The use of multi-camera networks to complete infrastructure inspections and drones to complete infrastructure monitoring is mentioned above. These encounter such problems as the number of cameras required, the relative position of the camera arrangement, planning the path of the drone, and selecting the shooting angle, which are all decision-making problems. For a multi-camera system, Hamdi et al. [145] developed a multi-view transformation network (MVTN) to recognize 3D shapes by automatically deploying the poses of six cameras through prior knowledge, and not simply at 60° intervals. Giordano et al. [146] proposed an indoor real-time multi-target tracking system based on a visual sensor network. To solve the problem of real-time 3D indoor structure monitoring and multi-target tracking, the system uses a distributed game theory algorithm to control the relevant parameters of the PTZ camera in order to ensure that it can maximize the coverage of the tracking area and solve the problem of high-accuracy recognition.

### 4.6. Sensor Fusion

As described in Section 3.2, laser–vision fusion enabled infrastructure monitoring with more dimensions, and higher accuracy, and more practicality compared to vision-based monitoring. However, there are some challenges that sensor fusion encountered, for example: (1) The combination of point laser and camera. Since the point laser is usually used as a rangefinder, its position relative to the camera needs to be carefully considered. For example, the design of IATS attempted to align the laser and camera centers to form a high-precision coaxial measurement system in Ref. [15]. The advantage of this fusion method is that the laser range values can be directly converted to image depth information, but the disadvantage is that the structure is very strict and difficult to maintain. Another common approach is to separate the laser and camera externally and to mount two sensors in fixed positions, employing a delicate mechanical structure. After calibration, they are placed in the same coordinate system. In Ref. [147], three 1-D laser sensors and a camera together formed a 6-DOF pose sensor system with a long measurement range, and the position of each sensor was defined precisely. These cases show that the fusion of point lasers should consider two aspects: mechanical design and calibration method. (2) The accuracy of LiDAR–camera fusion. As mentioned in Section 3.2.3, a majority of LiDAR–camera fusion methods are point-level fusions. However, the point-level fusion has two challenges: (1) compared with dense image pixels, LiDAR points are sparse. Using calibration matrices to find the hard association wastes many image features with rich semantic information; (2) the hard association heavily relies on image conditions and calibration quality. Bai et al. [148] proposed a soft association mechanism, enabling the network to adaptively determine where and what information should be taken from the images.

Based on the research described in the previous paragraph, we can summarize the future direction of vision–laser-based infrastructure monitoring: (1) replacing LiDAR with point laser sensors to design low-cost monitoring systems when the monitoring objective is predefined and the area is small; (2) improving the accuracy of LiDAR–camera monitoring. There are two issues related to monitoring accuracy that need further research: the association of sensors with different resolutions and sensor environment preference. The state-of-the-art fusion methodology requires abandoning the traditional association methods, which rely on the extrinsic calibration matrix, and then solving the problem of poor performance of LIDAR and the camera in harsh environments.

## 5. Summary and Conclusions

This article summarized the past and state-of-the-art research on non-contact infrastructure inspection and monitoring, which consists of two steps: vision-based inspection and vision–laser-based monitoring. The vision-based inspection started with different handcrafted methods to detect cracks, corrosion, and other defects, and then presented the limitations of this approach. Section 2.2 discussed the applications of machine learning-based object detection, which was divided into heuristic and deep learning-based detection, and then pointed out that object detection only aims to fit a rectangular box around the region of interest. The third major subsection reviewed the progress of semantic segmentation, which can delineate the precise location and shape of the damage. However, the lack of depth information prevented the transformation of image data into actionable information. To overcome this drawback, we presented infrastructure monitoring in Section 3.

The objective of monitoring is to acquire a quantitative understanding of the current state of the infrastructure. The third section first introduced the principle and application of only visual technology (DIC, MVS, and SFM) for monitoring infrastructure and then primarily discussed the limitations of a single sensor in the field of infrastructure monitoring, which led to the major content of Section 3: vision–laser fusion. The fusion technology can combine the advantages of these two sensors to achieve a highly accurate and efficient defect and target recognition. This article divided the sensor fusion technology into three parts based on the type of laser sensor: (1) visual single-point ranging; (2) line-structured light vision measurement; and (3) LiDAR visual infrastructure monitoring. In these three parts, the fusion principle, related applications of vision, and different laser technologies were conducted.

Besides an extensive literature review study, the most significant contribution of this article was discussing the challenges encountered by non-contact infrastructure inspection and monitoring, as well as future directions in Section 4. We believe that an automated non-contact infrastructure monitoring system begins with collecting raw data from different sensors in an outdoor scene and then reconstructs a 3D infrastructure model through big data management so that the defects can be located precisely. Finally, quantified defects will be used to evaluate structural consistency and safety. Therefore, future research works on this topic can aim at four areas: big data management, outdoor conditions, sensor fusion, and low-cost systems. The rapid advances in research in vision–laser-based inspection and monitoring of civil infrastructures described in this paper will enable accurate, time-efficient, cost-effective, automated, and non-contact civil infrastructure inspection and monitoring.

## Figures and Tables

**Figure 1 sensors-22-05882-f001:**
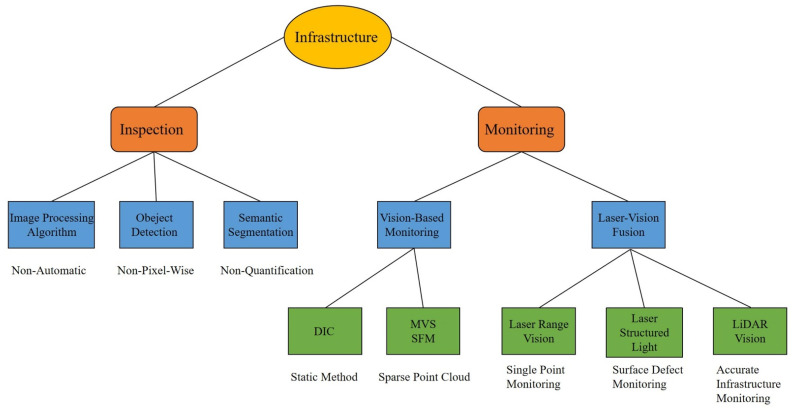
Workflow performed in this review paper.

**Figure 2 sensors-22-05882-f002:**
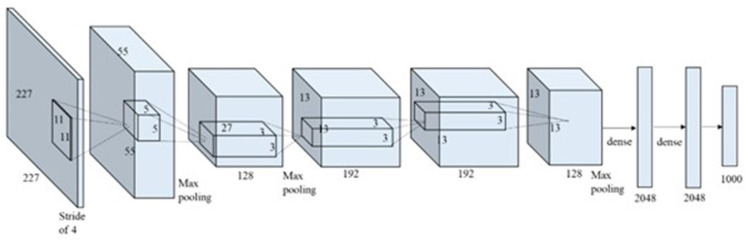
Overall architecture of AlexNet. Reprinted from Ref. [53] with permission of Sensors, 2018.

**Figure 3 sensors-22-05882-f003:**
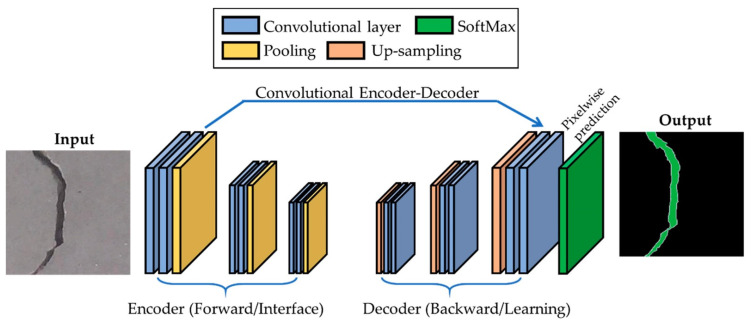
Block diagram of the proposed fully convolutional network (FCN) with the encoder–decoder framework. Reprinted from Ref. [60] with permission of Sensors, 2018.

**Figure 4 sensors-22-05882-f004:**
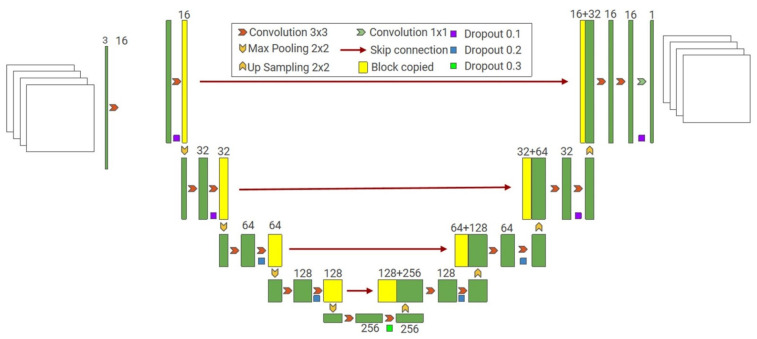
U-Net architecture.

**Figure 5 sensors-22-05882-f005:**
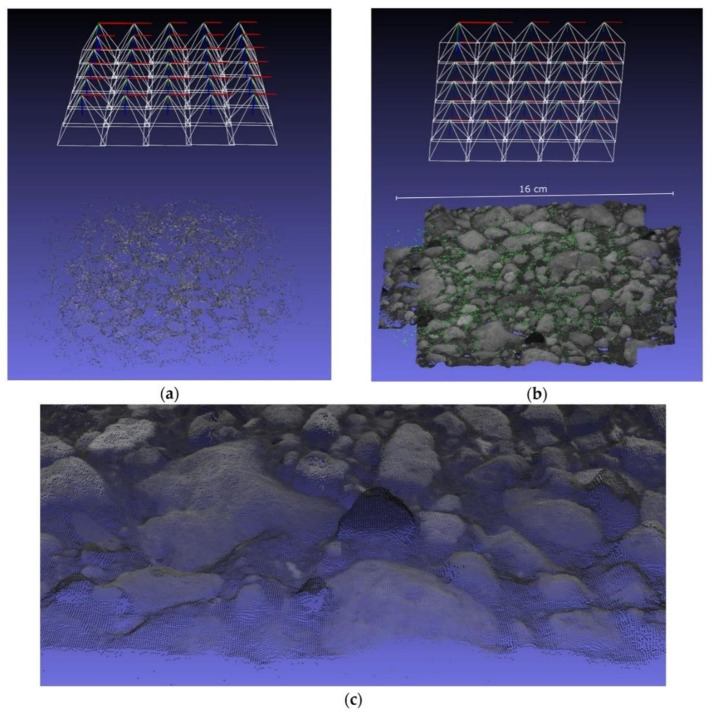
A 3D point cloud reconstruction of a particular concrete specimen: (**a**) Reconstructed camera poses including the sparse point cloud after SfM procedure, (**b**) generated dense point cloud after dense image matching (DIM), and (**c**) zoom-in of the dense point cloud. Reprinted from Ref. [89] with permission of Materials, 2020.

**Figure 6 sensors-22-05882-f006:**
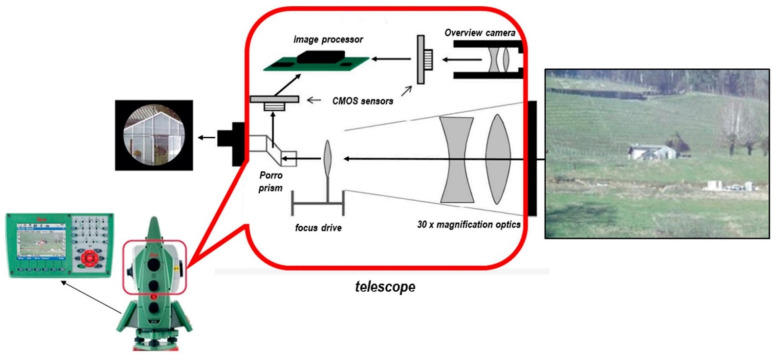
The image-assisted total station schematic cross-sectional view of the telescope with image sensors and processor integration, as well as motorization for autofocus; scheme modified for this study from [94].

**Figure 7 sensors-22-05882-f007:**
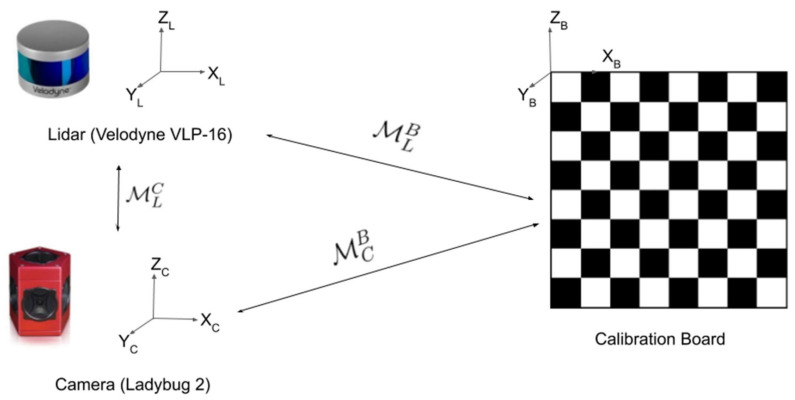
Principle of extrinsic calibration. The objective is to find the rigid transform between the LiDAR and the camera. It is currently mostly done manually using a calibration target, such as a 2D or 3D chessboard or pattern. Reprinted from Ref. [110] with permission of Sensors (Basel) 2020.

**Figure 8 sensors-22-05882-f008:**
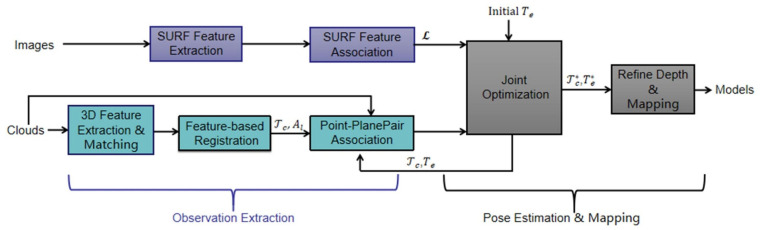
A diagram of the LiDAR–camera fusion method for accurate dense 3D reconstruction system pipeline; scheme modified for this study from [123].

**Figure 9 sensors-22-05882-f009:**
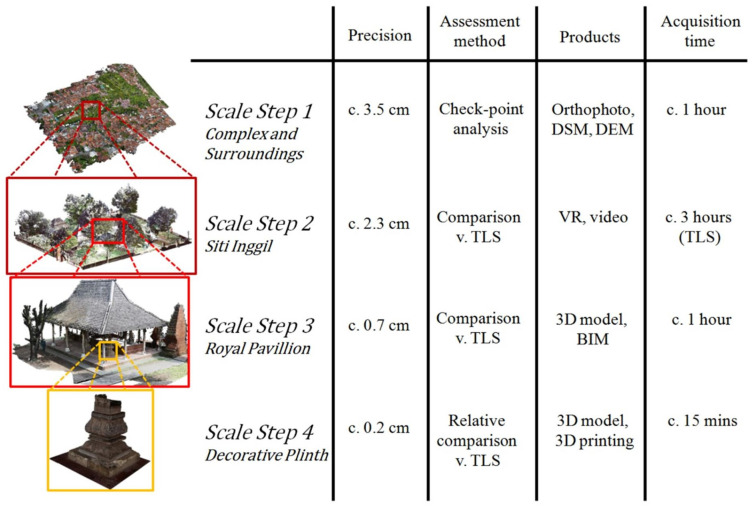
Example of the result of the multi-scale approach. In step 1, the whole Kasepuhan complex and its surroundings are modeled by aerial photogrammetry. Step 2 shows the Siti Inggil area within the palace compounds; here is shown the registered TLS point cloud. Step 3 shows a building, the Royal Pavilion, within Siti Inggil, which was modeled using close-range photogrammetry. Finally, step 4 shows an architectural detail, in this case a column’s plinth, also modeled using close-range photogrammetry. Reprinted from Ref. [129] with permission of the ISPRS International Journal of Geo-Information, 2018.

**Figure 10 sensors-22-05882-f010:**
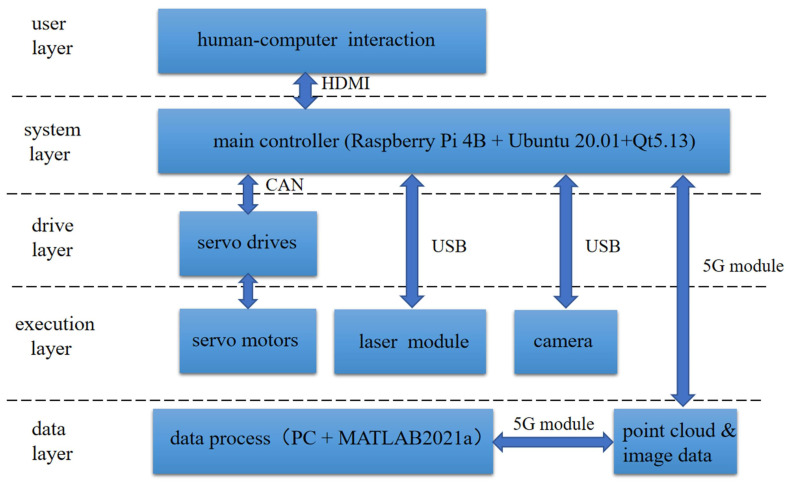
The frame of the indoor quality inspection system; scheme modified for this study from [144].

**Table 1 sensors-22-05882-t001:** Summary of image processing algorithms.

Defects Types	Ref.	Advantages
Crack	[22,23]	Detecting multi-scale cracks
	[25]	Comparing different binarization methods
	[24,26]	Threshold select, fast detection
	[27,28]	Remove noise (fog, rain, and shadow)
	[29,30]	Crack quantification
Corrosion	[31,32,33]	Fast color feature processing regardless of noise and illumination
Others	[34]	Railway defects detection
	[35]	Pavement pits detection

**Table 2 sensors-22-05882-t002:** Summary of object detection.

	Algorithms	Ref.	Results
Non Neural Network	SVM	[37]	Nine classes of the fastener, 98%
	Comparing Algorithms	[38]	STM fastener defects detection, 99.4%
		[39]	pavement crack, KNN > boosted tree > Recursive partitioning > bootstrap forest > linear regression > naive Bayes
Neural Network	Shallow NN	[40,41]	Boltzmann crack identification, 90.95%
	Deep NN	[42]	Deep NN process time-domain signal, 87%
		[47,48]	CNN-based defects detection, 98%
		[53,54]	Different size sliding window combination
		[55]	Faster RCNN detects different defects
Comparing Algorithms		[49,50,51,52]	Comparing CNN, SURF-based, NB-CNN, LBP-SVM, SVM, boosting, logistic regression, random forest, KNN

**Table 3 sensors-22-05882-t003:** Summary of semantic segmentation.

Semantic Segmentation Network	Ref.	Advantages
CNN-based	[56]	CNN with ResNet23 and VGG19_reduced
	[60]	VGGNet with decoder
FCN	[57,58]	FCN, deconvolution up-sample
	[59,61]	end-to-end semantic segmentation
U-Net	[63,64]	Textured-surface defects
	[61]	U-Net with Faster RCNN
	[66]	3D semantic segmentation
	[69]	Semi-supervised segmentation, 83.21%
Seg-net	[68]	Coordinate pooling

**Table 4 sensors-22-05882-t004:** Summary of vision-based infrastructure monitoring.

Measurement Algorithms	Ref.	Measurement Types	Disadvantages
DIC	[71,72,73,74]	2D-DIC	Strict experimental layout and measurement environment
	[75,76,77,78]	3D-DIC	
MVS	[81]	Using landmarks	Landmarks disposal
	[82,83]	SIFT-based measurement	Not accurate
SFM	[84,85,86,87]	SIFT-based 3D reconstruction	Time-consuming
	[88,89]	SURF-based monitoring	Not accurate
	[90,91,92]	Comparing SFM and Laser scanner	Laser scanner more accurate but time-consuming

**Table 5 sensors-22-05882-t005:** Summary of vision–laser-based infrastructure monitoring.

Fusion Methods	Ref.	Monitoring Types
Vision Range Laser	[95,96,97]	Total station-based deformation measurement
	[98]	Low-temperature environment deformation monitoring
	[99]	Railway crack detection
Structured Light Vision	[100]	Point laser structured light
	[101,102,103]	Texture surface monitoring
	[104]	Railway tunnels monitoring
LiDAR Vision	[111,112,113,114]	LiDAR camera calibration
	[115]	Infrastructure deformation
	[116,117]	Crack monitoring
	[118,119]	Pavement pit monitoring
	[120,121]	Surface defects monitoring with color information
	[122]	Subway obstacles and vehicles
	[123]	Large structures monitoring
	[124,125,126]	UAV with LiDAR and cameras
	[127,128,129]	Post-earthquake and urban area monitoring

## Data Availability

No new data were created or analyzed in this study. Data sharing is not applicable to this article.

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
