# Peer review of "A Review of Vision-Laser-Based Civil Infrastructure Inspection and Monitoring"

_sensors, 2022, doi:10.3390/s22155882_

Round 1

Reviewer 1 Report

The present manuscript is a thorough literature review of the the following field: Vision-Laser-Based Civil Infrastructure Inspection and Monitoring. It was a very pleasant reading and, as it stands, it is a key contribution to the general effort of upkeep of our aging infrastructure stock.

Some small improvements are as follows:

1. The reviewer strongly suggest a thorough check of the English. Some small examples are: "[...] are mostly performed manually" in the abstract or "installed on the infrastructure"  in the introduction.

2. Whenever stating "[...] Although the measurement accuracy is high, they cannot complete the target detection task alone", what is actually missing to complete said task? Specify.

3. "In conclusion, a single sensor can have a certain role; however, it has limitations. Therefore, in practical applications, these sensors should be combined to utilize their different functions." This sentence is unclear though important. Also, please remove "in conclusion" if this is not a conclusion of a section or manuscript.

4. In the introduction there are two separate section attempting to explain the content of the manuscript. They should be combined into one.

5. The manuscript is generally well structured. The only weak point is the distinction between inspections and monitoring. According to Figure 1, the review relies heavily on said distinction, yet this one is not clearly explained. Is there any moment where these two are defined? Especially in regards to the difference in tools used, the outputs, the post-processing tools, etc.? The reviewer feels like the paragraph between Section 2 and Section 2.1 could make a much better job at explaining how visual based inspections are performed and why section 2.1 immediately discusses the post-processing algorithm instead of the in-situ practice itself.

These are not to be taken like grievous lackings as, outside of these, the manuscript is well written and fully deserving to be published.

Author Response

We gratefully thank you for your constructive comments and suggestions, which has significantly imporved the quality of the manuscript. We have carefully considered each suggested revisions and comment of reviewer and make some changes.

The red part that has been revised according to your comments. Revision notes, point-to-point, are given as follows:

Point 1. The reviewer strongly suggest a thorough check of the English. Some small examples are: "[...] are mostly performed manually" in the abstract or "installed on the infrastructure"  in the introduction.

Response 1: Thank you so much for your careful check. We have checked entire manuscript.

Point 2. Whenever stating "[...] Although the measurement accuracy is high, they cannot complete the target detection task alone", what is actually missing to complete said task? Specify.

Response 2: Thank you for your comment. We have added the description of this part (line 6-10, page 2).

The revised phrase is [“Single-beam lasers are mostly used to detect translations of a single or a few points in a single direction. Although the measurement accuracy is high, they cannot complete the target detection task alone. For example, a single-beam laser rangefinder can perform single-point ranging to obtain depth information through ToF technology, but due to the lack of 2D plane information and color information, it cannot locate defects and cannot reflect defect properties such as color, area, shape, etc.”]

Point 3. "In conclusion, a single sensor can have a certain role; however, it has limitations. Therefore, in practical applications, these sensors should be combined to utilize their different functions." This sentence is unclear though important. Also, please remove "in conclusion" if this is not a conclusion of a section or manuscript.

Response 3: We gratefully appreciate for your valuable suggestion. We have replaced “in conclusion “ and revised this sentence to make it clearer (line 21-24, page2).

The revised sentence is [“As mentioned in the previous paragraph, a single sensor can have a certain role but also has limitations: monocular camera can perform defect inspection in a 2-dimensional plane, but the lack of depth information makes it impossible to quantify defects; single-beam laser cannot identify the entire object and locate the defects; LiDAR-based monitoring is accurate, but it is expensive, time-consuming and lacks color information.”]

Point 4. In the introduction there are two separate section attempting to explain the content of the manuscript. They should be combined into one.

Response 4: Thank you for your reminder. We have combined into one section (line 52-54, page 2).

The revised sentence is [“The fourth part of the article discusses the challenges of non-contact infrastructure inspection and monitoring. Correspondingly, Section 4 also presents future research and practical engineering applications toward these challenges.”]

Point 5. The manuscript is generally well structured. The only weak point is the distinction between inspections and monitoring. According to Figure 1, the review relies heavily on said distinction, yet this one is not clearly explained. Is there any moment where these two are defined? Especially in regards to the difference in tools used, the outputs, the post-processing tools, etc.? The reviewer feels like the paragraph between Section 2 and Section 2.1 could make a much better job at explaining how visual based inspections are performed and why section 2.1 immediately discusses the post-processing algorithm instead of the in-situ practice itself.

Response 5: We greatly appreciate yours constructive comments. We have added the desciption of distinction between inspections and monitoring (line 31-35, page 1), the description is [“There are some significant differences between inspection and monitoring: inspection focuses on the identification of surface defects in construction materials and structural defects in infrastructure, which is mostly realized by optical cameras and object recognition algorithms; monitoring obtains a quantitative understanding of the current state of the infrastructure by measuring physical quantities such as defect size, vibration frequency, accelerations and/or displacement, which is mostly a multi-sensor system.”]

Then we have rewritten the paragraph between Section 2 and Section 2.1 (line 7-19, page3).

The revised paragraph is [“Infrastructure inspection is an application of object detection technology in the field of construction. Relevant data are collected through various types of sensors, such as images, laser signals, and ultrasonic signals, and these signals are further processed to separate different characteristics to determine the state of the infrastructure. In recent years, the continuous development of computer vision technology has enabled the application of visual inspection in various engineering fields. Recent research on automatic vision-based inspection is generally divided into two steps: (1) acquiring image data by setting up static cameras or flying drones [18,19]; (2) processing image data by computer vision technology to complete defect and structure inspection. There are some challenges in data acquisition: (1) single static camera is limited by its field of view. (2) the flight path of camera-equipped drones heavily affects inspection results. These challenges are discussed in Chapter 3 and Chapter 4. Chapter 2 focuses on data processing techniques and introduces vision-based infrastructure inspection from three perspectives. The first part presents the types of most defects and different heuristics methods for damage detection using image data, and the other sections review the application of artificial intelligence technology in infrastructure inspection, which is divided into two parts: object detection and semantic segmentation.”]

The main revision in the paper and the response to the reviewers' comments are indicated in the attached file “Responses to comments”, and all changes are also highlighted in the attached Word file of revised manuscript.

We hope you will find our revised manuscript acceptable for publication.

Yours sincerely,

Chongwen Xu

Reviewer 2 Report

The paper provides an overview of vision-based inspection and vision-laser-based monitoring techniques and applications. It is a review paper which covers suitably and in a detailed way the state of the art of the topic. Therefore, it can be accepted 

Author Response

We appreciate your comments. According to the comments of all reviewers, we have made following major revisions:

  1. We have revised the introduction to make it clearer.
  2. We have added few photos and references to highlight the major contents of this manuscript.
  3. We have added one subsection of Section 4 to discuss the challenges of sensor fusion.
  4. We have rewritten the conclusion to emphasize the contributions of this manuscript.

All changes are also highlighted in the attached Word file of revised manuscript.

We hope you will find our revised manuscript acceptable for publication.

Yours sincerely,

Chongwen Xu.

Reviewer 3 Report

This paper conducted a review of vision-based inspection and vision-laser-based monitoring techniques for structural health monitoring. However, in-depth review and future direction from the review are lacking.

Author Response

We appreciate your comments. To make the review more in-depth and the future direction of infrastructure inspection and monitoring clearer. We have made following changes:

  1. We have revised the introduction to make it clearer.
  2. We have added few photos and references to highlight the major contents of this manuscript.
  3. We have added one subsection of Section 4 to discuss the challenges of sensor fusion.
  4. We have rewritten the conclusion to emphasize the contributions of this manuscript, then we have summarized the ongoing works which was described in Section 4 in 4 directions: big data management, outdoor conditions, sensors fusion, and low-cost system

All changes are also highlighted in the attached Word file of revised manuscript.

We hope you will find our revised manuscript acceptable for publication.

Yours sincerely,

Chongwen Xu.

Reviewer 4 Report

1. The number of keywords should be reduced. Please refine it.

2. Please improve the logic of Introduction. It should declare the background, progress and movitation of this manuscript.

3. Please improve the logci of the major contents. For different non-contact techniques,  the detail device and the construction of the system should be carefully described. Some photos should be given. Please add this content.

4.  The major content should be highlighted. Various methods and the data analysis methods have been expressed. However, it misleads the readers. Please focus on the major topic.

5. The conclusion part should be revised. It should give the major contribution of this review. What readers can learn from it.

6. Please check the reference 35.

Author Response

We gratefully thank you for your constructive comments and suggestions, which has significantly imporved the quality of the manuscript. We have carefully considered each suggested revisions and comment of reviewer and make some changes.

The red part that has been revised according to your comments. Revision notes, point-to-point, are given as follows:

Point 1. The number of keywords should be reduced. Please refine it.

Response 1: Thank you for your reminder. We have reduced the keywords to 4 (line 20, page1).

The revised Keyword is [“Keywords: infrastructure inspection and monitoring; non-contact; vision-laser-based; sensor fusion.”]

Point 2. Please improve the logic of Introduction. It should declare the background, progress and movitation of this manuscript.

Response 2: We greatly appreciate yours constructive comments. We have revised the introduction of the manuscript to make it clearer (line 21, page 1-line 54, page 2). And the new introduction was divided into 4 paragraghs: first paragraph is the background of this manuscript; second paragraph discusses the motivation and different types of non-contact sensors; third paragraph discusses the purpose of using sensor fusion technology in infrastructure monitoring; the last paragraph shows the progress of entire article.

Point 3. Please improve the logci of the major contents. For different non-contact techniques,  the detail device and the construction of the system should be carefully described. Some photos should be given. Please add this content.

Response 3: Thank you for your comment. We have revised the introduction and conclusion to improve the logic of major contents. And we also added 5 photos to describe the detail device and the construction of the system of the major contents. (semantic segmentation, vision-laser fusion, challenges of infrastructure inspection and monitoring)

Point 4.  The major content should be highlighted. Various methods and the data analysis methods have been expressed. However, it misleads the readers. Please focus on the major topic.

Response 4: We gratefully thanks for the precious time the reviewer spent making constructive remarks. We have added the bulleted list in the last few lines of the introduction (line 55, page2-line 3, page 3), it shows below:

  • Machine vision-based infrastructure inspection, especially semantic segmentation;
  • Infrastructure monitoring, quantitative understanding of the current state of the infrastructure;
  • Vision-Laser fusion technologies and their applications.
  • The challenges and ongoing works toward automated non-contact infrastructure inspection and monitoring;

And the final section of the manuscript (line 3-29, page 18) also highlighted the major contents of the manuscript.

Point 5. The conclusion part should be revised. It should give the major contribution of this review. What readers can learn from it.

Response 5: We gratefully appreciate for your valuable suggestion. We have rewritten entire Section 5 (Summary and Conclusions). The revised conclusion not only describes contributions of this review but also shows our discussions for the future direction of non-contact infrastructure inspection and monitoring (line 3-29, page 18).

Point 6. Please check the reference 35.

Response 6: Thank you for your reminder. We have have adjusted the format of reference 35.

The main revision in the paper and the response to the reviewers' comments are indicated in the attached file “Responses to comments”, and all changes are also highlighted in the attached Word file of revised manuscript.

We hope you will find our revised manuscript acceptable for publication.

Yours sincerely,

Chongwen Xu

Round 2

Reviewer 3 Report

While the authors have improved the manuscript, the further direction of the development of the vision laser based structural health monitoring techniques is still unclear after this review as most of the techniques presented by the authors are known so far.  

Author Response

We gratefully thank you for your constructive comments and suggestions, which has significantly imporved the quality of the manuscript. We have carefully considered each suggested revisions and comment of reviewer and make some changes.

The red part that has been revised according to your comments. Revision notes, point-to-point, are given as follows:

Point 1. While the authors have improved the manuscript, the further direction of the development of the vision laser based structural health monitoring techniques is still unclear after this review as most of the techniques presented by the authors are known so far. 

Response 1: We greatly appreciate yours constructive comments. We have added a paragraph in section 4.6 to make the future direction of vision-laser-based monitoring clearler (line 3-9, page 19). It shows below:

Based on the research described in the previous paragraph, we can summarize the future direction of vision-laser-based infrastructure monitoring: (1) replacing LiDAR with point laser sensors to design low-cost monitoring systems when the monitoring objective is predefined and the area is small; (2) improving the accuracy of LiDAR-camera monitoring. There are two issues related to monitoring accuracy that need further research: the association of sensors with different resolutions and sensor environment preference. The state-of-the-art fusion methodology requires abandoning the traditional association methods which rely on the extrinsic calibration matrix and then solving the problem of poor performance of LIDAR and the camera in harsh environments.

The main revision in the paper and the response to the reviewers' comments are indicated in the attached file “Responses to comments”, and all changes are also highlighted in the attached Word file of revised manuscript.

We hope you will find our revised manuscript acceptable for publication.

Yours sincerely,

Chongwen Xu

Reviewer 4 Report

The authors made corrections according to the comments.

Please declare the limitation of using contacted sensors (i.e., optical fiber sensor) for structural health monitoring, so as to highlight the advantages of vision-laser based inspection and monitoring.

Please consider to get information from the two references.

Improving the durability of the optical fiber sensor based on strain transfer analysis. Optical Fiber Technology, 2018, 42, 97-104. Priority design parameters of industrialized optical fiber sensors in civil engineering. Optics and Laser Technology, 2018, 100, 119-128.

Author Response

We gratefully thank you for your constructive comments and suggestions, which has significantly imporved the quality of the manuscript. We have carefully considered each suggested revisions and comment of reviewer and make some changes.

The red part that has been revised according to your comments. Revision notes, point-to-point, are given as follows:

Point 1. Please declare the limitation of using contacted sensors (i.e., optical fiber sensor) for structural health monitoring, so as to highlight the advantages of vision-laser based inspection and monitoring.

Please consider to get information from the two references.

Improving the durability of the optical fiber sensor based on strain transfer analysis. Optical Fiber Technology, 2018, 42, 97-104. Priority design parameters of industrialized optical fiber sensors in civil engineering. Optics and Laser Technology, 2018, 100, 119-128.

Response 1: Thank you for your reminder. We have added the example of contact sensors (line 42-43, page 1)

For example, optical fiber sensors used to measure the strain and temperature information in civil infrastructure require packaging technique and embedded installation [3,4].

  1. Wang, H.; Jiang, L.; Xiang, P. Improving the durability of the optical fiber sensor based on strain transfer analysis. Optical Fiber Technology 2018, 42, 97-104.
  2. Wang, H.; Jiang, L.; Xiang, P. Priority design parameters of industrialized optical fiber sensors in civil engineering. Optics & Laser Technology 2018, 100, 119-128.

The main revision in the paper and the response to the reviewers' comments are indicated in the attached file “Responses to comments”, and all changes are also highlighted in the attached Word file of revised manuscript.

We hope you will find our revised manuscript acceptable for publication.

Yours sincerely,

Chongwen Xu
